SOFTWARE

# ntStat: k-mer characterization using occurrence statistics in raw sequencing data

**Parham Kazemi**[1,2]*, **Lauren Coombe**[1], **René L. Warren**[1], **Inanc Birol**[1,3]*

**1** BC Cancer Research Institute, Vancouver, Canada, **2** Bioinformatics Graduate Program, Faculty of Science, University of British Columbia, Vancouver, Canada, **3** Department of Medical Genetics, University of British Columbia, Vancouver, Canada

\* pkazemi@bccrc.ca (PK); ibirol@bccrc.ca, inanc.birol@ubc.ca (IB)

## Abstract

*K*-mer counts are fundamental in many genomic data analysis tasks, providing valuable information for genome assembly, error correction, and variant detection. State-of-the-art *k*-mer counting tools employ various techniques, such as parallelism, probabilistic data structures, and disk utilization, to efficiently extract *k*-mer frequencies from large datasets. The distribution of *k*-mer counts in raw sequencing reads reveals key genomic characteristics such as genome size, heterozygosity, and base-calling quality. The number of reads containing a *k*-mer has also shown application in genome assembly and sequence analysis. We present ntStat, a toolkit that employs succinct Bloom filter data structures to track both *k*-mer count and depth information and use in downstream applications. ntStat models the *k*-mer count histogram using evolutionary computation, and infers valuable insights about the genome, sequencing data, and individual *k*-mers, *de novo*. ntStat consistently ran faster than DSK, BFCounter, hackgap, and Squeakr in all of our tests. Jellyfish performed faster than ntStat for human data with k = 25 but fell behind with k = 64. KMC3 was faster overall but at a high disk usage and memory cost. ntStat also used less memory than other non-disk-based k-mer counters and typically, 99.5-99.9% of the k-mers processed by ntStat are counted correctly. ntStat's histogram analysis module detected heterozygosity percentages and k-mer coverage for long-read datasets simulated from a diploid human genome with less than 1% and 0.5-fold difference to the ground truth. The analysis of simulated long read datasets showed an average error of just 2% in k-mer robustness estimates.

## Author summary

Characterizing an organism's genome from raw sequencing data is a foundational step in biological research, informing disease studies and evolutionary analysis. This process often relies on counting short DNA substrings, i.e., k-mers, to estimate genome size and complexity, and yet, the massive scale of modern

**Data availability statement:** ntStat is freely available at github.com/BirolLab/ntStat.

**Funding:** This study is supported by the Canadian Institutes of Health Research (CIHR) [PJT-183608, I.B.] and the Natural Sciences and Engineering Research Council of Canada (NSERC) [I.B.]. None of the authors received a personal salary award from any of the funders; staff salaries (for L.C. and R.L.W.) and student stipends (for P.K.) were paid by the host institution using research funds from these grants. The funders had no role in study design, data collection and analysis, decision to publish, or preparation of the manuscript.

**Competing interests:** The authors have declared that no competing interests exist.

datasets frequently creates computational bottlenecks when counting k-mers. We present ntStat, a toolkit that overcomes these challenges by using succinct data structures to rapidly gather k-mer statistics. In addition, ntStat employs evolutionary computation to model k-mer count distributions and infers, *de novo*, various characteristics for individual k-mers, including heterozygosity and coverage. Our benchmarks show that ntStat consistently outperforms state-of-the-art tools in speed and memory efficiency, maintaining over 99.5% k-mer count accuracy while eliminating the need for massive disk storage and memory. By lowering these computational barriers, ntStat provides a scalable and accessible solution for extracting biological insights.

## 1 Introduction

The recent growth in abundance of genome sequencing data has been key to unlocking insights into genetic variation, gene expression, and evolution. However, this surge in data volume has introduced many computational challenges, highlighting the need for scalable genomic data analysis solutions. Many of these solutions utilize k-mers, substrings of length k, as a core data type for performing tasks [1] including indexing, sketching, and constructing De Bruijn graphs [2]. Specifically, k-mer count statistics have proven to provide simple and effective information for a variety of applications, such as GeneToCN [3] which leverages gene-specific k-mer counts to estimate gene copy numbers, and STRling [4] where k-mer frequency data is used to call novel short tandem repeat expansions. In addition to k-mers, spaced seeds (or gapped k-mers), where predetermined positions in the k-mers are ignored, are valued for their sensitivity in homology search [5].

The efficacy of k-mer-based tools depends heavily on the availability of efficient and accurate methods for extracting k-mer count statistics and characterizing vast genomic datasets. Despite the simplicity of naïve k-mer counting algorithms, this task becomes computationally intensive for large datasets. Several k-mer counters, such as DSK and KMC3 [6] scale to large datasets by utilizing computer disk storage. ntHits [7], BFCounter [8], and Squeakr [9] employ compressed data structures, such as Bloom filters [10] and counting quotient filters [11], to manage memory use efficiently. Jellyfish [12] improves counting speed by using a lock-free hash table.

The capability to filter k-mers based on counts, e.g., removing erroneous k-mers or finding duplications using highly repetitive k-mers, is also useful in various applications. For example, genome assembly polishers, such as ntEdit [13] and JASPER [14], improve assembly quality by selecting k-mers with frequencies above a set threshold. While DSK and KMC allow for minimum and maximum abundance thresholds, BFCounter only outputs k-mers appearing at least twice, and Jellyfish, Squeakr, and ntHits lack maximum count thresholds altogether. Moreover, while hackgap [15] is the only tool that supports spaced seeds, it does not allow for the simultaneous filtering of k-mers based on both minimum and maximum count thresholds. Therefore, there is a need for a versatile k-mer and spaced seed counting tool that can filter

*k*-mers based on specified minimum and maximum frequency thresholds, without requiring a large amount of disk space and while maintaining speed and memory efficiency.

Utilizing *k*-mer frequencies for sequence analysis in genomics is conceptually similar to the bag-of-words model in natural language processing (NLP). Both models represent sequential data by counting occurrences of smaller units, *k*-mers in genomic sequences and words in text, while disregarding the order. Although the bag-of-words model is effective for basic NLP tasks such as text classification on small datasets, more sophisticated methods like term frequency-inverse document frequency (TF-IDF) have shown to improve results in more complex applications [16]. TF-IDF adjusts word frequencies by considering their prevalence across a collection of documents, assigning greater weight to informative yet less-common words. This concept can be applied to genomic data, treating sequencing reads as individual documents and normalizing *k*-mer frequencies according to the number of reads covering each *k*-mer. For example, Canu [17], a long-read genome assembly tool, avoids collapsing repeats and haplotypes by using TF-IDF information. Additionally, Tiara [18] leverages *k*-mer TF-IDFs in a deep neural network to classify eukaryotic sequences in metagenomic data. These tools necessitate algorithms that can extract the required TF-IDF statistics efficiently, which are not captured by any of the existing state-of-the-art *k*-mer counters.

While *k*-mer frequencies have shown utility in bioinformatics applications, the histogram of *k*-mer counts alone uncovers many characteristics of the sequencing data and underlying genome, including data quality and genome heterozygosity. This histogram can be efficiently generated without keeping track of all individual *k*-mers. ntCard [19] and KmerEstimate [20], for instance, count a subsample of *k*-mers and estimate true frequencies using statistical models. Typically, *k*-mer count histograms are a mixture of components representing *k*-mers likely generated by sequencing errors, and those accounting for genome ploidy and repetitive regions. GenomeScope [21,22] extracts genomic characteristics by modelling *k*-mer count histograms from short-read sequencing data and separating the components corresponding to erroneous, heterozygous, and duplicated *k*-mers. With the increasing prevalence of third-generation sequencing platforms, such as the ones offered by Oxford Nanopore Technologies (ONT), Plc. (Oxford, UK) and Pacific Biosciences (PacBio), Inc. (Menlo Park, USA), it is crucial to adapt *k*-mer count histogram models to account for higher error rates and sequencing biases, as these factors impact genome profiling accuracy.

Here, we introduce ntStat, a toolkit designed for efficient *k*-mer and spaced seed frequency extraction and statistical analysis. ntStat streamlines *k*-mer count extraction while filtering *k*-mers with frequencies between set thresholds. Our tool also incorporates flexible statistical methods to characterize *k*-mers, making it a valuable addition to the existing toolkit landscape.

## 2 Design and implementation

ntStat features two primary modules. The counting module obtains *k*-mer occurrence frequencies and depth information using Bloom filters, and the histogram analysis module approximates a mixture model to allow the inference of *k*-mer statistics from their counts. ntStat's implementation is modular, where each component is programmed in C++ as a pybind11 module (38) or in pure Python.

### 2.1 Count and depth extraction

ntStat's count and depth extraction module operates by inserting *k*-mers into multiple Bloom filters (BFs) and counting Bloom filters (CBFs). A BF is a space-efficient probabilistic data structure that uses a set of independent hash functions to map a *k*-mer to specific indices in a bit array. During insertion, the bits at these indices are set to true. To query for membership, the corresponding indices are checked to all be set to true, indicating that the *k*-mer is likely present. A CBF extends this concept by storing integer counters at these indices; to query a count, the CBF reports the minimum count stored in the hashed indices (Fig 1a). As depicted in Fig 1b, the global BF stores all distinct *k*-mers from the dataset. If a *k*-mer occurs more than once, it will be inserted in the intermediate CBF. Typically, *k*-mer count histograms show that

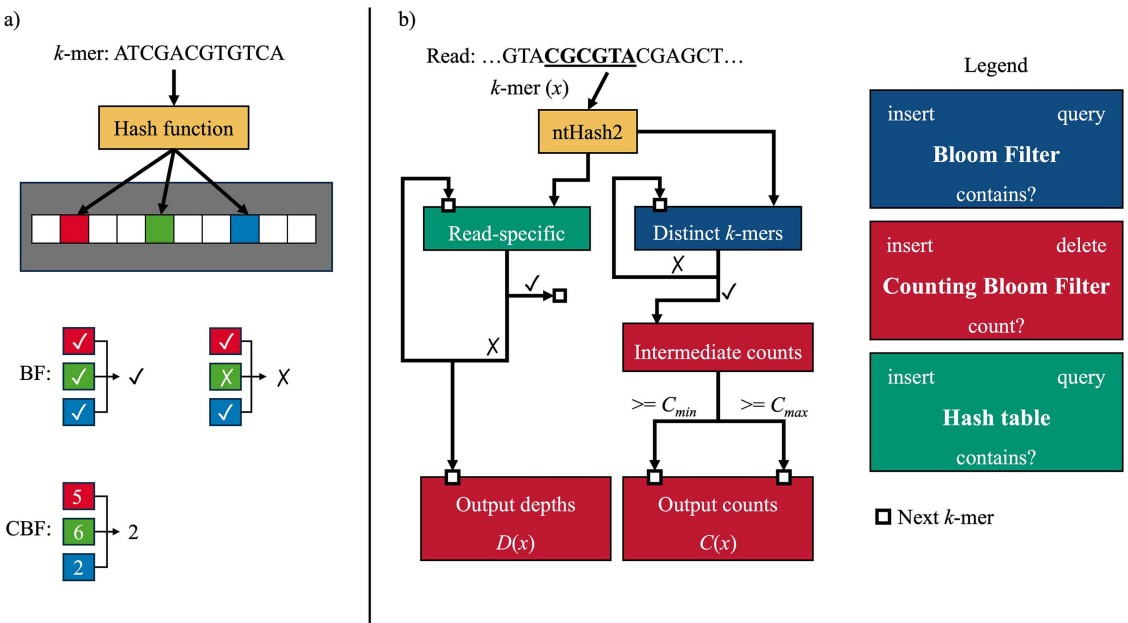

**Fig 1. a) schematic of BF and CBF query operations b) layers of Bloom filters used in ntStat for concurrently finding k-mer count and depth information.**

most $k$-mers occur only once in sequencing data. Hence, the intermediate CBF holds a small subset of $k$-mers. Since CBFs require a byte to be allocated for each hash value, as opposed to one bit in BFs, storing less elements in the CBF substantially reduces memory requirements. Finally, $k$-mers that are observed more than a user-defined threshold $C_{min}$ are inserted in the output CBF. In cases where count information is not required and $k$-mer membership queries are sufficient, ntStat can output a BF instead, which requires 8x less space. If a $k$-mer occurs more than another threshold $C_{max}$, it will be removed from the CBF or stored in a secondary output BF. This functionality is facilitated by the decrement function implemented in CBFs available in the btllib library [23]. This implementation of CBFs also employs atomic counters to allow multithreaded execution. ntStat only allocates memory for necessary BFs, depending on the set parameters. For instance, when $C_{min} = 2$, the CBF for storing intermediate counts is not constructed, further reducing memory usage.

One of ntStat's novel features is the ability to extract read-specific $k$-mer frequency information, simultaneously while gathering global $k$-mer counts. To compute the number of reads containing a given $k$-mer, which we refer to as the $k$-mer's "depth," ntStat constructs read-specific BFs large enough to contain all $k$-mers in a read with a negligible false-positive rate. While processing the $k$-mers within a read, ntStat increments the $k$-mer depth in a global CBF only if the $k$-mer has not been previously added to the read-specific table. The resulting count and depth information is stored in CBFs, which as a compressed data structure, requires less storage compared to plain text files. Users can integrate these BFs or CBFs into their own applications via the btllib package, or leverage the utilities provided by ntStat to export the counts in TSV format.

## 2.2 Histogram analysis

The main objective of the histogram module is to model the probability that individual k-mers are erroneous or originate from heterozygous or homozygous regions. The input to this module is a $k$-mer profile generated by ntCard [19] and denoted as $(h_1, h_2, \ldots, h_c)$, where $h_i$ represents the number of distinct elements that appear $i$ times in the sequencing data, and $c$ is the maximum $k$-mer count in the histogram. We utilize ntCard to generate this input because it efficiently

subsamples the input data to estimate the histogram without the overhead of exhaustively tracking every *k*-mer count. This optimizes performance and enables rapid analysis where *k*-mer counting is not required.

*K*-mer count histograms commonly follow a mixture model $F(x) = \sum_{i=1}^{p} w_i f_i(x)$, where $p$ is the ploidy of the genome and $w_i$ and $f_i$ represent the weights and probability density functions of the $i^{th}$ component. The components' distributions vary depending on the genome and sequencing platform. Unlike previous studies in the literature, which often assume the distribution of the error component and peaks, ntStat does not impose any predefined assumptions. Instead, it dynamically selects and combines distributions from a pool, including gamma and exponential for the error [21,22,24], and normal, skew normal, Poisson, and negative binomial for the genomic components [25,26], making it versatile for analyzing data from various sequencing platforms.

This flexibility is achieved by using evolutionary algorithms to search for the most suitable model. Specifically, we use Differential Evolution (DE) [27], a robust global optimization method particularly well-suited for problems with a large search space and a computationally inexpensive objective function. We define the error of a model as the sum of absolute differences between its predicted values and the input histogram, normalized as a probability distribution:

$$Err(F) = \sum_{x=1}^{c} |F(x) - \frac{h_x}{\sum_{j=1}^{c} h_j}|$$

(1)

Candidate solutions are represented by models that receive the distribution types of the erroneous *k*-mer counts and two major histogram peaks along with the three corresponding parameters sets and component weights. DE explores a wide range of possible solutions by iteratively evolving a population of candidates. In each iteration, new candidate solutions are generated through recombination and mutation, followed by selection based on error. Ultimately, the solutions in the population converge towards a global optimum. After the DE algorithm terminates, ntStat attempts to further improve the solution's distribution parameters using a local search technique—the Broyden–Fletcher–Goldfarb–Shanno algorithm [28–31].

ntStat reports the final solution to the user and provides options for plotting the model, visualizing solutions found in each iteration of the optimization process as an animation. The relative area under the curve values of the model's components is used to estimate overall base quality, *k*-mer coverage, and genome heterozygosity (Table A in S1 Text). In conjunction with the count module, the histogram model's components' relative probabilities can be used to characterize individual *k*-mers (Section 3.4).

## 3 Results

We evaluated the *k*-mer counting and histogram analysis functions on datasets from multiple species and platforms, as well as on simulated data with known attributes. We also compared ntStat's scalability, efficiency, and accuracy against state-of-the-art *k*-mer counting tools.

### 3.1 Counting accuracy

To analyze the factors affecting the accuracy of ntStat's output *k*-mer counts and its performance, we simulated Illumina short reads with 30-fold coverage and 0.1% error rate from the *E. coli* genome [32] using pIRS [33] and generated the count histogram ($k = 25$) using ntCard (Fig 2a). We created this dataset to facilitate the evaluation process, since generating ground-truth *k*-mer counts and validating all of the *k*-mer counts is a compute-intensive process. The counting module's performance is independent from the underlying genome's nature and sequencing technology.

As demonstrated in Fig 2b, there is a trade-off between accuracy and memory usage as the error parameter is adjusted. Increasing the error rate reduces memory consumption, while also decreasing accuracy. The main factor affecting the memory usage and output accuracy of ntStat's counting module is the size of the BFs used in the cascade.

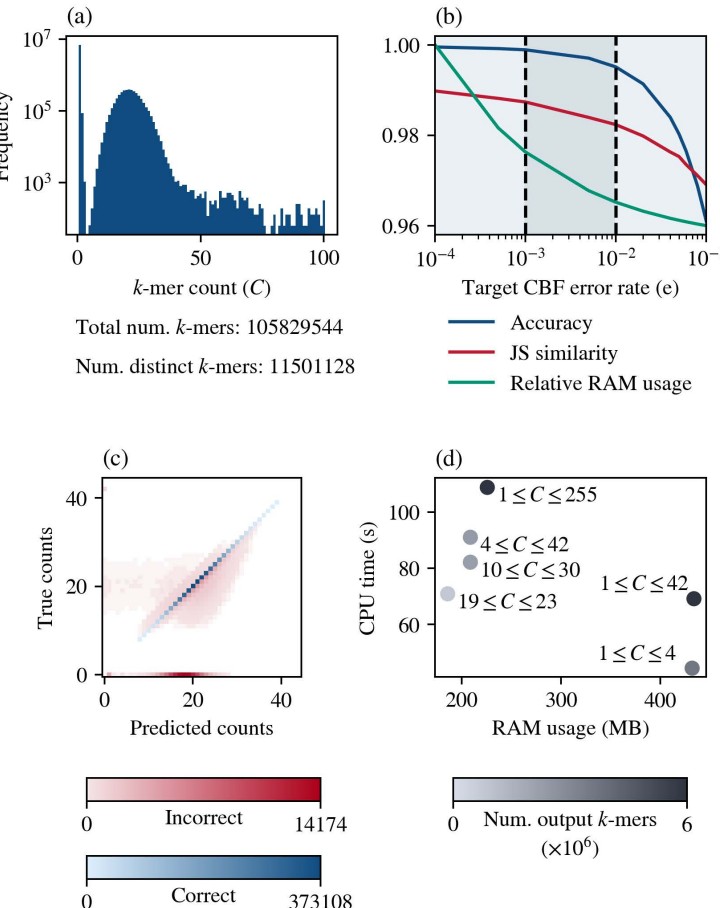

**Fig 2. Impact of ntStat's arguments on k-mer counting accuracy and performance. (a)** Count histogram of the simulated test data in log scale. **(b)** Changes in count accuracy, i.e., the ratio of counts correctly reported, and memory requirements when tuning the error rate parameter ($C_{min}=4$, $C_{max}=42$). Histogram similarity is calculated as the Jensen-Shannon similarity between the histogram generated from the output CBF and the ground truth histogram. The knees of the accuracy and RAM usage curves are shown as dashed lines. RAM usage was scaled to a range of 0.96 to 1, where 1 corresponds to 970MB. **(c)** Count error distribution for $C_{min}=4$, $C_{max}=42$, and $e=0.1$. Color intensities represent the number of correct and incorrect output k-mer counts. **(d)** CPU time and memory usage of different count slices ($e=0.01$). Color intensities represent the number of k-mers in each slice.

Users can control this by either specifying a fixed output BF size or allowing ntStat to calculate the optimum sizes by setting the error rate parameter (-e, 0.001 by default, enabling 99.5-99.9% k-mer count accuracy) and providing the count histogram. In this case, the desired error rate along with the fixed number of hash functions (seven hashes per k-mer) and the number of k-mers expected to be inserted in each BF/CBF will determine their sizes, according to the Bloom filter false-positive rate formula. As shown in Fig 2c, errors can be due to false positives in the output CBF, where the true count is expected to be zero, but ntStat reports non-zero values. In addition, false positives in the cascade's BF and CBF can cause minor deviations from true counts.

Along with the desired error rate, the count thresholds also impact ntStat's performance. In Fig 2d, we demonstrate how changing the count thresholds leads to variations in memory usage and CPU time. Generally, setting thresholds where the area under the count histogram for that range is larger require higher processing times, since more k-mers will pass through multiple Bloom filters in the cascade. As mentioned earlier, ntStat optimizes memory usage and speed

by avoiding memory allocation to redundant Bloom filters, such as when $C_{min} = 2$, where the intermediate CBF is skipped. Similarly, the output counting Bloom filter is the only data structure created when the counts are unbounded, i.e., $C_{min} = 1$ and $C_{max} = 255$. As 8-bit unsigned integers are the default data type used in the CBF counters, $C_{max} = 255$ is the largest possible value for the upper threshold.

## 3.2 Counting speed and resource usage

We evaluated ntStat's (v1.0.0) counting speed and efficiency in comparison to DSK (v2.3.3), KMC (v3.2.4), Jellyfish (v2.3.0), BFCounter (v0.2), Squeakr (v0.7) and hackgap (v1.0.1), using sequencing datasets from the *C. elegans* and *H. sapiens* genomes (Table B in S1 Text). Each experiment was set to run with 48 parallel threads (Fig A in S1 Text). For DSK, no multiplicity limitations were set for the output k-mers. Only *k*-mers that occurred more than once were counted by BFCounter and Squeakr, and the minimum count threshold was set to 3 occurrences for hackgap. All other parameters of the tools were set to their defaults. We disabled ntStat's depth extraction functionality to ensure a fair comparison. On average, we observed that enabling this module increases memory requirements and run time by 27.4% and 5.2% respectively, with the same threshold and target error rate parameters set for counting. The majority of the increased memory requirement is due to the CBF containing depth information, as read-specific BFs are substantially smaller and are deallocated once the read has been processed.

As illustrated in Fig 3a, 3b, and 3c), ntStat took 2.1m to process the *C. elegans* dataset, roughly 1.6x, 26x, 21x, and 7x faster than DSK, BFCounter, Squeakr, and hackgap, respectively. ntStat ran in 49.0m on average for the *H. sapiens* dataset with *k* = 25, 1.1x, 4.9x, 12.8x, and 5.7x faster than the same comparators. With *k* = 64, ntStat again ran in 51.8m, whereas DSK, BFCounter, and Squeakr took 2.3h, 7.5h, and 15.8h, respectively. Hackgap is not included in this

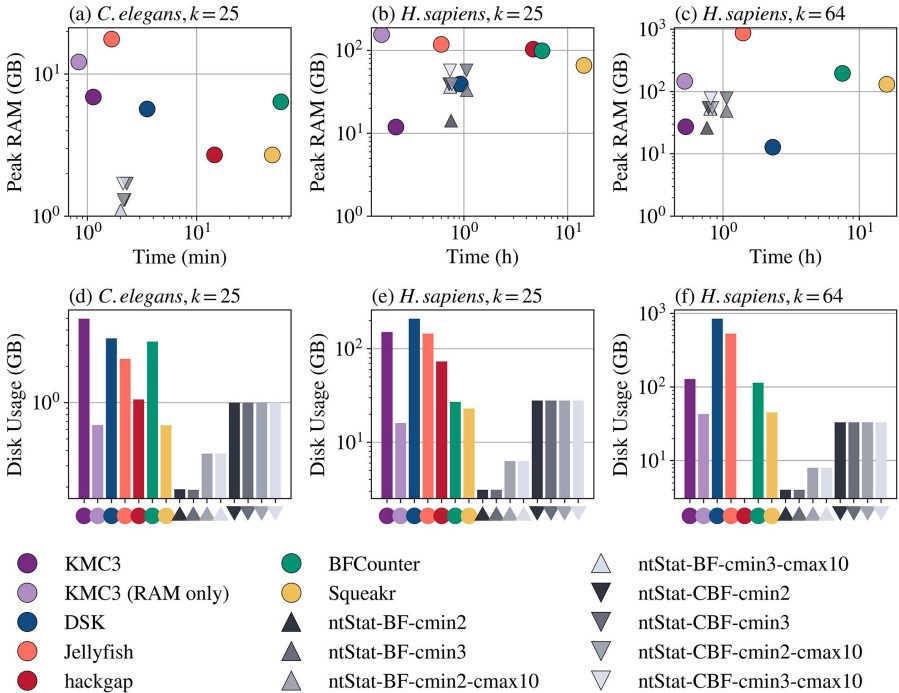

**Fig 3. Speed and memory usage of ntStat's counting module compared to other methods. a, b, c)** Time and memory usage (in log-log scale) of various ntStat configurations compared with other tools for three datasets. 'BF' and 'CBF' refer to the output data type. **d, e, f)** Total output file size (GB, log scale) generated by the tools.

experiment, since it did not support $k=64$ and threw a runtime error. Jellyfish is 1.2x and 1.3x faster than ntStat for *C. elegans* and human with $k=25$, and 1.6x slower for $k=64$, due to the scalability of ntHash2 to larger $k$-mer sizes. KMC3 ran faster than ntStat (3.7x for $k=25$ and 1.6x for $k=64$), but at the cost of substantial disk usage during run time (5GB for *C. elegans*, 151.0GB for human with $k=25$, and 128.0GB for human with $k=64$). In addition, KMC3 consumed 5.9x the memory used by ntStat for *C. elegans* data on average, in default mode.

ntStat required 0.3-1.7GB of RAM for *C. elegans* data, vs. 5.7GB, 17.6GB, 6.4GB, 2.7GB, and 2.7GB for DSK, Jellyfish, BFCounter, Squeakr, and hackgap, respectively. KMC3 allocated 6.9GB of RAM for *C. elegans* with its default configuration, and 12.2GB in its RAM-only mode. As for the human data, ntStat used 26.2-78.2GB and 14.2-57.3GB RAM for $k=64$ and $k=25$ respectively, vs. 27.3GB and 11.9GB for KMC3 in default mode, and 145.9GB and 155.5GB in RAM-only mode. The memory usage of ntStat was 60–73% less than BFCounter and Squeakr. Jellyfish used 876.0GB memory for $k=64$ and 118.3GB for $k=25$, i.e., more than 11x and 2x that of ntStat for the two $k$-mer sizes, respectively. ntStat's memory usage was comparable to that of DSK for *H. sapiens* with $k=25$, and 2x with $k=64$, without using any disk space during execution.

Fig 3d, 3e, and 3f) also demonstrates the disk space efficiency of the Bloom filters saved by ntStat, which were 0.03-0.3x the size of DSK's HDF5 files, and up to 71% smaller than BFCounter's hash tables. It took 1.6s to query $10^7$ random 25-mers from the CBF generated by ntStat from *C. elegans* data ($C_{min}=2$), 14s for Squeakr, 5.9s for KMC3, and 3 minutes for Jellyfish.

Since hackgap is the only comparable tool capable of counting spaced seeds, we ran the *C. elegans* experiment with gapped seeds and visualized the results in Fig B in S1 Text. To ensure a fair assessment of memory efficiency, we manually configured hackgap's intermediate data structure sizes to match the capacity of ntStat's intermediate Bloom filters, since hackgap requires the user to indicate the sizes of these data structures. We observed that ntStat maintains consistent performance across varying $k$-mer sizes and error rates, with run times 4.9–5.8x faster than hackgap for various seed patterns at $k=25$ and $k=30$.

### 3.3 Histogram modelling analysis

To evaluate the accuracy of the insights reported by the histogram analysis module, we simulated sequencing reads from the *H. sapiens* (3.05Gbp), *C. elegans* (100.2Mbp) and *D. melanogaster* (137.5Mbp) reference genomes using pIRS [33] for short reads, and NanoSim [34] for ONT, respectively. Given a haploid reference genome, we created a copy for the second haplotype, incorporating various sequence variation (e.g., single nucleotide variants, short indels, structural variants) rates to model allelic heterozygosity. We obtained the $k$-mer count histograms for each dataset using ntCard and then analyzed them with ntStat's histogram module. Furthermore, we labelled $k$-mers from the simulated reads present in both the reference and copied assemblies as "robust", and $k$-mers present exclusively in one of the simulated haplotypes as heterozygous, to form the ground truth expected to be detected by ntStat (Table C in S1 Text).

First, we created two different copies of the haploid *H. sapiens* genome assembly with low (0.1%, i.e., copy 1) and high (1%, i.e., copy 2) single nucleotide variation (SNV) rates, and simulated ONT-like reads using NanoSim from the assembly and both copies separately. According to Fig 4a and 4b, ntStat can detect the heterozygosity rates of both copies from the fitted model with less than 1% error, compared to the ground truth, showing its capability at resolving haplotypes. In these two cases, ntStat fitted the exponential and negative binomial distributions for the erroneous and genomic peaks, respectively.

Next, to show ntStat's performance on datasets from different sequencing technologies, we extended our experiments to the *D. melanogaster* genome and one copy haplotype with 1% SNV rate. As shown in Fig 4c, simulating ONT reads from this genome created a dataset with high coverage (193.8x) and a histogram with well-separated components. While the error component for this dataset was modelled with an exponential distribution, the genomic peaks were fitted by normal distributions. This is due to the central limit theorem, where a negative binomial distribution with a large shape

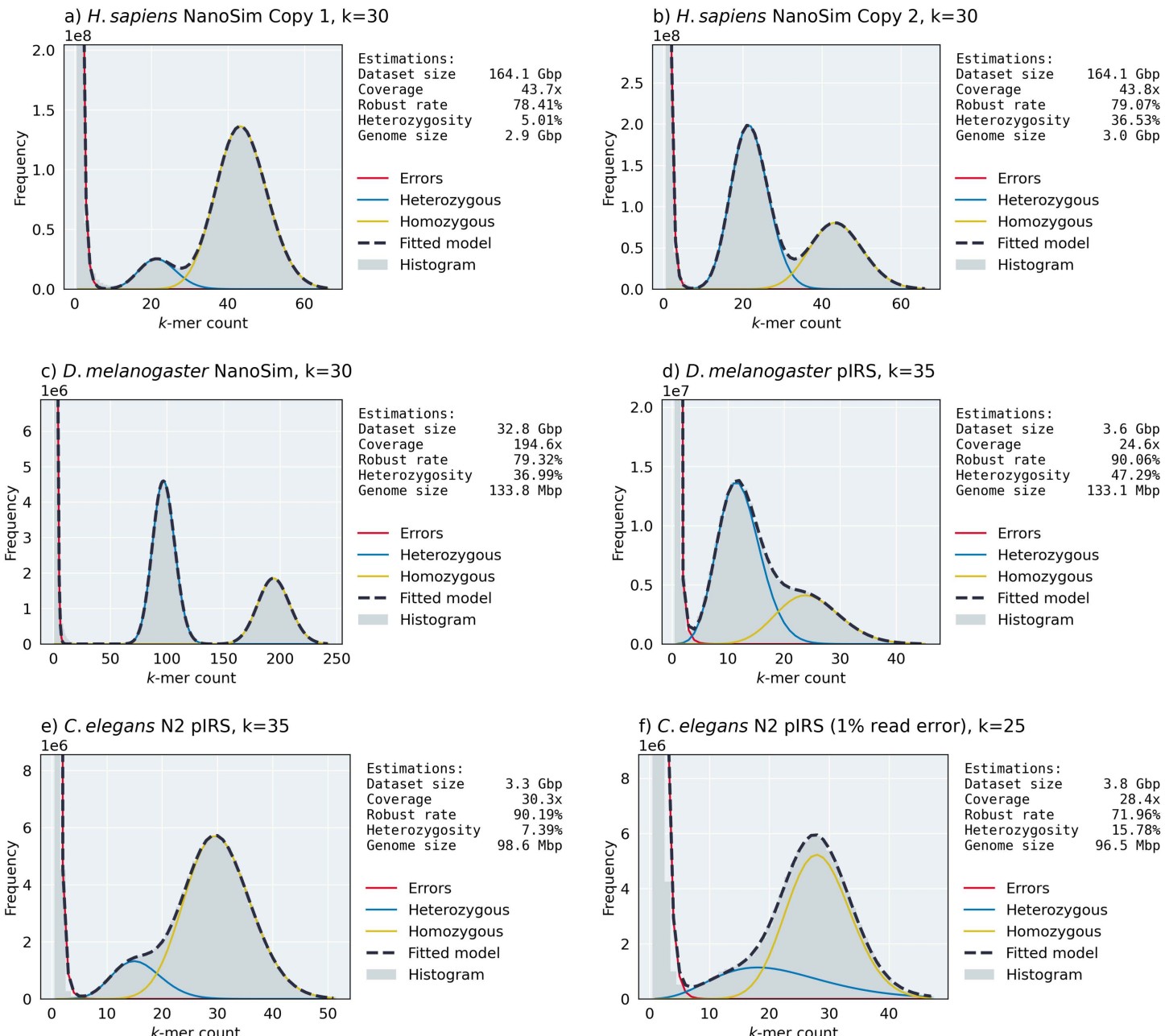

**Fig 4. Examples of ntStat's histogram analysis module's output on various simulated datasets.** We chose $k=30$ for moderately erroneous Nano-Sim reads, $k=35$ for the less erroneous pIRS-simulated reads, and $k=25$ for the more erroneous pIRS-simulated reads.

parameter converges to a discrete normal. Similar to the simulated human datasets, ntStat was able to approximate the fold-coverage with 0.5x error and a genome size of 134Mbp (3.5Mbp error). We find that ntStat estimates the rate of robust $k$-mers in datasets generated with NanoSim with 2% error, on average. The lower-coverage Illumina reads from the *D. melanogaster* genome, lead to less-separable genomic peaks (Fig 4d). Although the final model's mean absolute error is higher for this case compared to the NanoSim-generated data, ntStat inferred the coverage, robust $k$-mer rate,

and heterozygosity with 0.7x, 1.48%, and 0.85% error, respectively. For short read data, the difference between the total number of $k$-mers and dataset size is larger than for long-read data. Each sequencing read contains $L - k + 1$ $k$-mers, where $L$ is the read length, resulting in $k - 1$ bases per-read being ignored when calculating the dataset size from the total number of $k$-mers. This discrepancy systematically affects the accuracy of the genome size estimated by ntStat for short read data, which lead to 3.6% error in computing the length of the *D. melanogaster* genome.

Finally, we simulated two short-read datasets from the *C. elegans* genome (0.1% copy SNV rate) with 0.1% and 1% base error rates. Fig 4e and 4f show the robustness of ntStat's estimations regardless of the error rate. Specifically, ntStat estimated the coverage, robust $k$-mer rate, and heterozygosity of the more erroneous read set (26.33% error $k$-mers) with 0.1x, 1.71%, and 0.45% difference, respectively. ntStat modelled all of the histograms generated from pIRS-simulated datasets as a mixture of exponential and negative binomial components.

To characterize the performance of ntStat's histogram module compared to established profiling tools, we evaluated its output against GenomeScope using simulated datasets (Tables C and D in S1 Text). The results show the distinction in the intended scope of the tools. While GenomeScope is optimized for estimating global genomic properties, such as overall ploidy, genome size, and heterozygosity, ntStat is designed to model the underlying probability of individual $k$-mers belonging to error, heterozygous, or homozygous components. For instance, in the simulated *H. sapiens* dataset, ntStat accurately predicted that 5.01% of the total $k$-mer population originated from heterozygous regions (ground truth 5.05%), providing the granular probabilities necessary for classifying specific $k$-mers. In contrast, GenomeScope correctly reported the global nucleotide heterozygosity rate of 0.21%, confirming that while both tools analyze $k$-mer frequency histograms, ntStat provides the specific $k$-mer-level resolution required for downstream tasks such as error filtering, variant calling, and genome assembly polishing, while GenomeScope provides biological insights into the genome.

We also experimented with the histogram analysis module on experimental long and short read sequencing data (Table E in S1 Text). As shown in Fig 5, ntStat fits accurate models to the histograms generated from ONT (Fig 5a), PacBio (Fig 5b), and Illumina (Fig 5c) sequencing technologies. ntStat modelled all three histograms in this experiment with gamma and negative binomial components. Moreover, the reported 2.9Gbp-long human genomes align with the size of the complete T2T human genome assembly CHM13 v1.1 [35].

The speed of the histogram analysis module mostly depends on the population size and mutation and recombination rates set for the DE algorithm. These hyperparameters balance the exploration of the search space and exploitation of information gained from previous iterations during optimization. More exploration may lead to more accurate models, with the caveat of requiring more iterations to reach convergence. Regardless, our tests show that ntStat takes 2-12s to converge after ~50–300 iterations using the default parameters (population size of 16, mutation rate varying between 0.5 and 1.0 in each generation, and 0.8 recombination rate).

### 3.4 *K*-mer characterization

The outputs from ntStat's two main components provide valuable insights for analyzing genomic data at the k-mer level. By leveraging the histogram model's probabilities, $k$-mer counts, and depth information, we can effectively characterize $k$-mers. For this experiment, we simulated 500,000 ONT reads with quality scores using NanoSim, based on chromosome 7 from the HG002 diploid T2T assembly [36]. We collected the counts and depths of all k-mers within the reads and modeled their histogram (Fig 6a). The posterior probability of a $k$-mer count $x$ arising from the error, heterozygous, or homozygous component $c$ is computed as $P(\text{Component} = c \mid x) = w_c f_c(x)/F(x)$, where $f_c(x)$ is the probability of the component and $F(x) = w_{err} f_{err}(x) + w_{het} f_{het}(x) + w_{hom} f_{hom}(x)$ is the full model (Section 2.2 and Fig 6b).

We randomly sampled 100,000 distinct k-mers from the reads and labeled each based on the component with the highest posterior probability in the model and its frequency in the dataset. To assign ground truth labels, we determined the presence of each k-mer in neither, one, or both chromosome copies, categorizing them as erroneous, heterozygous, or homozygous, respectively. As shown in the confusion matrix (Fig 6c), the histogram model, combined with the counts in

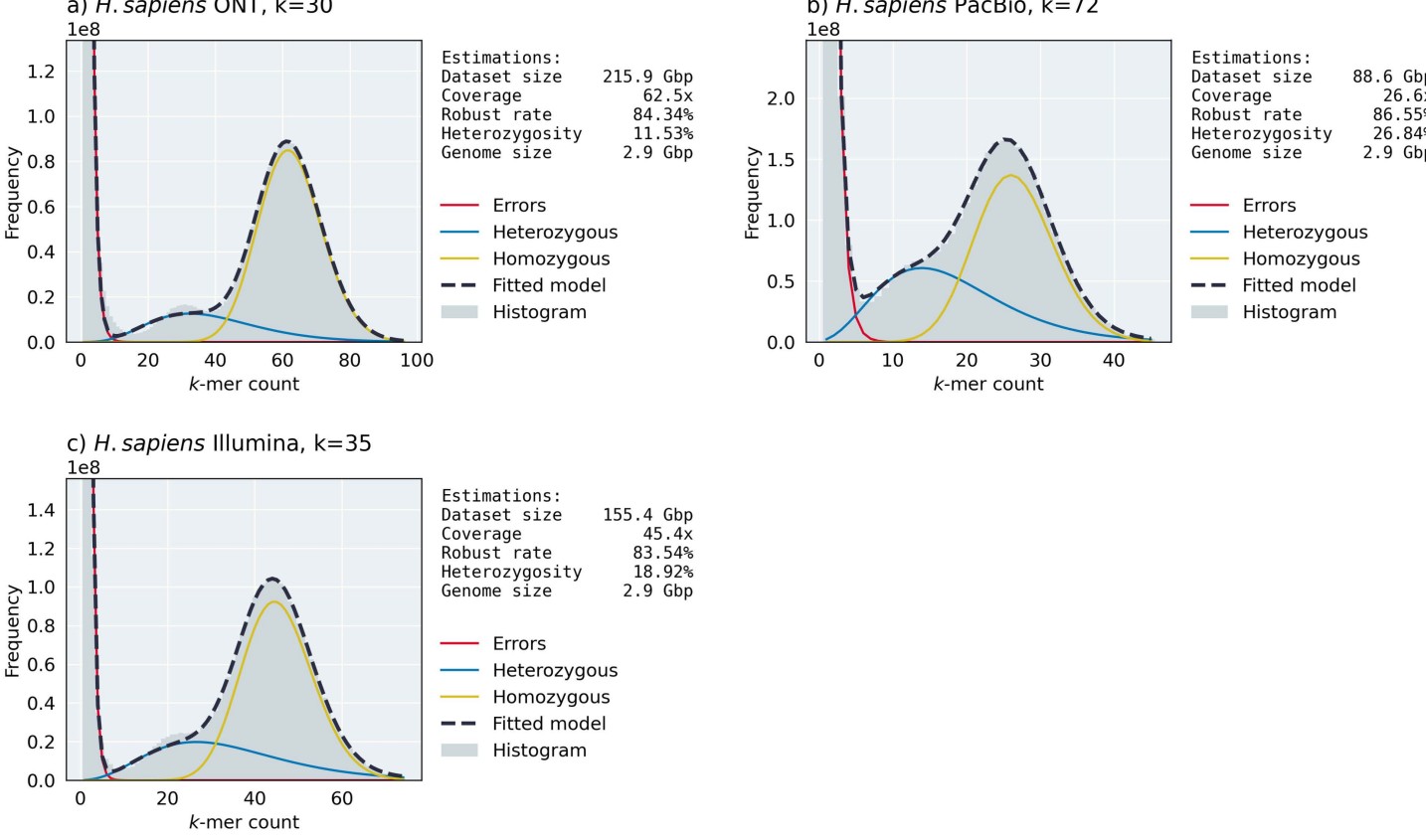

**Fig 5. ntStat's output generated from experimental sequencing data.**

the CBF, accurately detects erroneous *k*-mers and labels heterozygous and homozygous *k*-mers with accuracies of 75% and 99%, respectively.

We also examined the relationship between the minimum Phred score of the bases in each sampled *k*-mer and the probability of error reported by ntStat. A single base error within a *k*-mer designates it as erroneous, making the minimum base quality a representative score for the *k*-mer. As illustrated in Fig 6d, the distribution of Phred values recorded for *k*-mers with a given frequency indicates that the posterior probability of the error component can serve as a proxy for the Phred score.

To illustrate the utility of *k*-mer depth, we draw a parallel to the TF-IDF model in natural language processing, where the count-to-depth ratio serves as a genomic analogue to the term frequency-inverse document frequency statistic. In our model, count represents term frequency and depth represents document frequency. We analyzed the depths of all *k*-mers present in chromosome 7 of the HG002 reference genome. As shown in Fig 6e, the majority of *k*-mers exhibit equal count and depth values. However, *k*-mers with lower depth than count are those that occur multiple times within at least one read. This phenomenon is more common in data generated by long-read sequencing platforms. Additionally, when multiple occurrences of a *k*-mer are observed in close proximity, e.g., in *k*-mers that cover tandem repeats, they are more likely to be captured contiguously within a single read. Higher count-to-depth ratios in *k*-mers indicate that these occurrences are situated closer together in the genome. Notably, depth serves as a more reliable indicator of close-proximity repetitive *k*-mers compared to count. As shown in Fig 6f, higher count-to-depth ratios also suggest, with greater confidence, that the median distance between consecutive occurrences of the *k*-mer is shorter compared to *k*-mers with lower count-to-depth ratios.

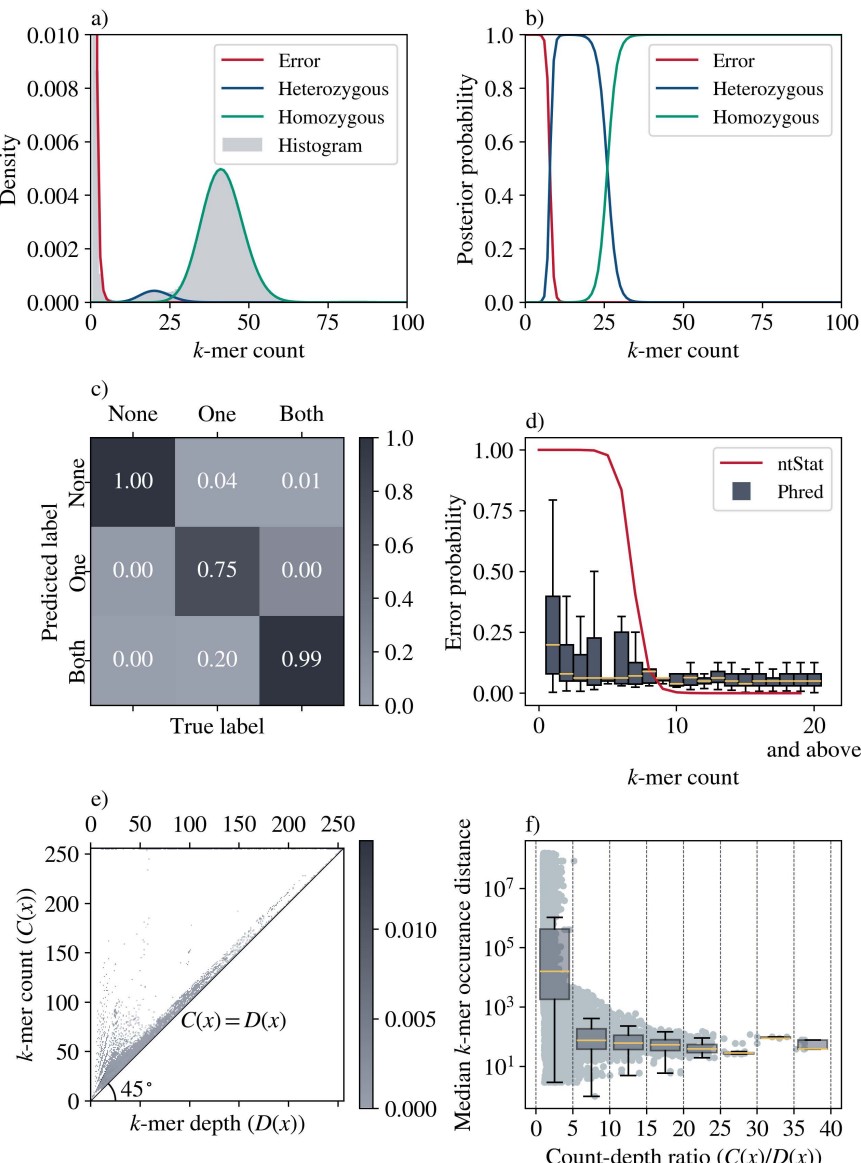

**Fig 6. Example application of ntStat's k-mer characterization module. a)** Model components fitted to the histogram of *k*-mer counts from ONT reads simulated from chromosome 7. **b)** Posterior probabilities of the components. **c)** Confusion matrix of the *k*-mer classification task based on presence in the chromosome copies. **d)** Relationship between *k*-mer error probabilities reported by ntStat and *k*-mer Phred scores, represented as probabilities. Phred score of Q is converted to an error probability with $P = 10\text{-}Q/10$. **e)** Heatmap of *k*-mer counts and depths. The color intensity of $(i, j)$ represents the density of *k*-mers with a count of $C(x) = i$ and depth of $D(x) = j$. **f)** Relationship between the count-to-depth ratio of a *k*-mer and the median distance between the *k*-mer's occurrences.

## 3.5 Discussion of results

Our experiments highlight the efficiency of ntStat's counting component, the potential benefits of incorporating k-mer depth information in genomic data analysis, and the accuracy of the biological insights gained by the histogram analysis module.

Overall, the counting module competes favourably in compute resource usage against state-of-the-art k-mer counting tools, especially ones that do not require disk usage to operate, such as Jellyfish, BFCounter, and Squeakr. ntStat's

output counts are stored in CBFs, efficient data structures that allow for faster queries than Squeakr, KMC, and Jellyfish. DSK generates a plain text file containing the k-mers and counts, which renders the querying process impractical due to the lack of indexing. As for BFCounter, there is no programming interface available to query the output hash table, forcing users to rely on a plain text file that presents similar limitations as DSK. The concept of k-mer depth parallels document frequency in NLP's TF-IDF model, where the depth of a k-mer is the number of sequencing reads containing that k-mer. One of ntStat's novel capabilities is the efficient calculation of k-mer depths using Bloom filters, a feature absent in all other k-mer counting tools.

In our controlled experiments on simulated long-read sequencing data, we found that ntStat's histogram analysis module reports heterozygosity percentages, k-mer robustness, and k-mer coverage with less than 1%, 2%, and 0.5-fold difference to the ground truth. This accuracy is due to the use of evolutionary algorithms for finding the suitable mixture of distributions and their respective parameters. ntStat uses the model to summarize basic information about the sequencing dataset. While GenomeScope provides more comprehensive biological insights into genomic features, such as heterozygosity, we anticipate that ntStat will primarily find utility in applications that require characterizing individual k-mers. For example, by combining k-mer counts and the histogram model's components, users can identify erroneous k-mers, which facilitates the development of robust k-mer-based algorithms [13,37], particularly given the higher error rates observed in long-read datasets compared to short-read ones.

## 4 Availability

ntStat is a novel tool for extracting k-mer TF-IDF statistics, modelling k-mer count histograms, and estimating genome, dataset, and k-mer characteristics de novo. The ability to extract k-mer statistics is crucial for developing genomic data analysis pipelines, particularly in an era of rapid data growth. Combined with its balanced usage of computational resources to rapidly create succinct data structures with a small memory footprint, ntStat has the potential to accelerate bioinformatics applications that rely on k-mer counts. Overall, the insights provided by ntStat show high potential for supporting large-scale genome projects and enhancing downstream analyses. ntStat is freely available at github.com/BirolLab/ntStat.

## 5 Future directions

Future development of ntStat will focus on expanding its utility beyond genomic assemblies to include RNA-seq and metagenomic data analysis by tailoring the statistical components to address the specific noise profiles and dynamic ranges of these diverse data types. Because the underlying hashing engine, ntHash2, natively supports the RNA alphabet (A, C, G, U), the core counting functionality requires no modification to process transcriptomic sequences.

## Supporting information

**S1 Text. Fig A.** Wall clock time of ntStat's filter command with different arguments vs. the number of threads. The dataset consisted of reads simulated from the C. elegans reference genome and contained 2,135,806,380 k-mers (k = 30). **Fig B.** Wall clock time of hackgap compared to ntStat for different k-mer sizes and minimum count thresholds (cmin2 and cmin3). Spaced seed patterns "1110111100011100011110111" and "111011110001110011100011110111" are used for s25 and s30, respectively, and k25 and k30 represent k-mers with no spaced seed masking applied. **Table A.** List of statistics summarized using the histogram model. "Err" and "Peak" refer to the distributions selected for the erroneous and genomic peaks, respectively. The number of k-mers with count i are shown as hi, and c refers to the maximum k-mer count available in the histogram. Each component is parameterized by w and θ, and the final model is represented by f(x). **Table B.** Datasets used for benchmarking ntStat's counting module's performance. **Table C.** Specifications of the simulated datasets shown in Fig 4. 'Copy SNV rate' refers to the -s and -d parameters set when creating the second haplotype using pIRS. 'Number of robust k-mers' and 'number of heterozygous k-mers' are the total number of k-mer present in at least one and exactly

one of the haplotypes, respectively. Percentages of robust and heterozygous k-mers are relative to the total and robust k-mers, respectively. 'Robust coverage' is calculated as the number of robust k-mers divided by the total number of k-mers present in the initial assembly. The script we used for obtaining these ground truths is available on ntStat's GitHub repository. **Table D.** GenomeScope's output for the simulated datasets (S3 Table). **Table E.** Data accession numbers, number of iterations until convergence, and final model error for the histogram analysis experiments on real data. (DOCX)

## Author contributions

**Conceptualization:** Parham Kazemi, Lauren Coombe, René L. Warren, Inanc Birol.

**Formal analysis:** Parham Kazemi.

**Funding acquisition:** Inanc Birol.

**Methodology:** Parham Kazemi.

**Resources:** René L. Warren.

**Software:** Parham Kazemi, Lauren Coombe.

**Supervision:** Inanc Birol.

**Validation:** Lauren Coombe, René L. Warren.

**Visualization:** Parham Kazemi, Inanc Birol.

**Writing – original draft:** Parham Kazemi.

**Writing – review & editing:** Lauren Coombe, René L. Warren, Inanc Birol.

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
