## [Decision Letter · Decision Letter 0]

22 Sep 2025

ntStat: k-mer characterization using occurrence statistics in raw sequencing data

PLOS Computational Biology

Dear Dr. Kazemi,

Thank you for submitting your manuscript to PLOS Computational Biology. After careful consideration, we feel that it has merit but does not fully meet PLOS Computational Biology's publication criteria as it currently stands. Therefore, we invite you to submit a revised version of the manuscript that addresses the points raised during the review process.

Please submit your revised manuscript within 60 days Nov 22 2025 11:59PM. If you will need more time than this to complete your revisions, please reply to this message or contact the journal office at ploscompbiol@plos.org. Please include the following items when submitting your revised manuscript:

We look forward to receiving your revised manuscript.

Kind regards,

Michael Domaratzki

Academic Editor

PLOS Computational Biology

Nir Ben-Tal

Section Editor

PLOS Computational Biology

**Additional Editor Comments:**

The reviewers have each identified some important considerations and limitations of the work. I encourage the authors to thoroughly review the comments and consider the changes requested - comparisons to other work, the histogram module, etc. This would be important for the revision to make a paper that would be suitable for publication.

**Journal Requirements:**

2) Your manuscript is missing the following sections: Design and Implementation, and Availability and Future Directions. Please ensure that your article adheres to the standard Software article layout and order of Abstract, Introduction, Design and Implementation, Results, and Availability and Future Directions. For details on what each section should contain, see our Software article guidelines:

https://journals.plos.org/ploscompbiol/s/submission-guidelines#loc-software-submissions

**Reviewers' comments:**

Reviewer's Responses to Questions

Reviewer #1: This manuscript developed a new software called ntStat for counting k-mersfrom sequencing data. ntStat has two modules: the k-mer counting module and the histogram analysis module.

The k-mer counting module use bloom filters (BF) and counting bloom filters (CBF) for k-mer counting. It is important to understand the basic ideas of BF and CBF to understand how ntStat works. The basic idea of a bloom filter is that that it uses multiple hash functions for each k-mer, with each hash value corresponds to a bit in the output hash table. If one hash value of the a new k-mer is not set in the hash table, it means that this k-mer has not been counted before. For counting BF, we store the counts of hash values, instead of bool values, which means the count is incremented for each input hash value. The minimum value of the counts over all hash values of a k-mer is used as the count of the input k-mer. This means CBF can overestimate the k-mer count if there is a hash collision.

Due to the usage of bloom filters, ntStat can run faster with less memory, with the tradeoff of accuracy loss. In the results, section 3.3 and 3.4, authors have demonstrated that ntStat are either faster than SOTA methods or use less memeory. The novel features of ntStat are (1) it can filter k-mers (without sorting) whose count numbers are in a predefined range c_min and c_max; (2) it can calculate the k-mer depth information, e.g. how many reads contain a given k-mer. The k-mer depth information can be used to derive the frequency-inverse document frequency (TF-IDF), which are useful for genome assembly tools.

The histogram analysis module is more vaguely described. It seems that the ntStat tries to fit a 3-component mixture model to the k-mer count density plot, with the components representing errors, heterozygous and homozygous. However, the background biological problem is not clearly described and the model specifications are missing, as well as the inference algorithm. In section 3.5-3.6, authors try to illustrate that ntStat can relibly estimate haplotypes given the proportion of error, heterozygous and homozygous components from sequencing data with different configurations.

I think this is decent software paper if authors can address the following major concerns:

1. add a more detailed introduction to BF and CBF with toy examples in the methods

2. add a mathematical formulation of the histgram analysis. what is the research question? what is the mathematical model to address the research question, e.g. the parametric forms of the components? How the inference algorithm works, what is the objective function?

3. i do not get the connection between the k-mer counting module and the histogram analysis module. the input to the histogram analysis module are generated by ntCard, which I assume is a published software. i thought the input are generated by the k-mer counting module.

4. it was argued that TF-IDF is important, but I did not see authors using it in the applications of the results section. it will be nice to add an application that use this information.

5. clearly describe the contributions of ntStat. If the histogram analysis part is already known method, then the contribution of ntStat is a re-implementation or extension of GenomeScope.

6. can GenomeScope and ntStat be used to analyse the same data? if yes, please add a comparion of the results on the two software on the same data.

7. Supplmentary are not available

Minor comments:

1. Fig. 1, the workflow of the couning process is not obvious, try to add more text in the legend and improve the figure, e.g. explain "global" and "depth"

2. section 3.2, first sentence, add ref to ntCard

3. section 3.2, last paragraph, "ntStat reports .... and classifying k-mers". could you elaborate what is "classifying k-mers"?

4. Fig. 2b, add the absolute memory usage corresponding to 0.96 and 1.

5. section 3.3, last paragraph, "....where the area under the count .... for that range is larger", "for that range is larger" => "larger range"

6. section 3.5, first paragraph, "Furthermore, we counted .... and copy as robust", split the sentence into shorter ones and explain what is "copy as robust"

7. section 3.5, paragraph start with "Next, to show ...", "we find that ntStat estimate the rate of robust k-mers ......", what is "robust k-mers"?

8. section 3.6, first paragraph, "p(component = c | x)... the density of the component", F_c(x) is not density, should be "probability". Also, it seems to be more reasonable to define F_c(x) = w_c * f_c(x), where w_c is the prior weight of component c and f_c(x) is the likelihood of x given component c.

Reviewer #2: The paper introduces a new k-mer counting approach with attention to

memory efficiency. The same team already has previous work (ntCard) on

the same subject. Here, novel metrics such as frequency–inverse

document frequency and histogram fitting are presented.

Main remarks:

- The method appears incremental compared to established tools (e.g., KMC, Genoscope), which perform better in their specialized tasks.

Reported speed gains may not justify tradeoffs, and reduced disk usage

is less compelling given current storage availability (especially on

machines equipped with, e.g., many threads as presented in the

benchmark).

- I think it is nice to have an indexed output. However how do authors

feel about forcing a specific output format and data structure with

respect to adoption? For instance, many modern k-mer indexing methods emphasize cache-efficient designs.

- Similarly, the emphasis on C++ with Python bindings being "seamless"

is more a benefit for the authors’ development workflow than for

external users in my opinion. The tool has many dependencies and is not

especially easy to install.

- Results reported on 48 CPUs are unclear (CPUs vs cores vs threads),

could we see scalability curves instead?

- The need to specify Bloom filter size or error parameters is not

user-friendly; many users will not know how to provide these. Can the

authors justify their design here?

- The manuscript focuses on genomics, and the fitting models are

supposed to be adjustable, but the paper does not demonstrate utility

in RNA-seq or metagenomics, which could also limit adoption.

- The tool is claimed to be useful for long reads, but sequencing error

rates are improving for these technologies. To that regard, could you

specify ntStat's actual supported range of k values? (I'd like to

clarify if large k are supported)

- Key comparisons to GenomeScope are hidden in the supplement. From the text, GenomeScope appears to provide more comprehensive biological insights (heterozygosity, ploidy).

The specific added value of ntStat remains unclear. If ntStat is to

underpin error-correction methods (as suggested in the paragraph before

the conclusion), runtime for recovering erroneous k-mers should be addressed.

Minor :

- succinct probabilistic Bloom filters : there's no Bloom filter

that is not probabilistic. "Succinct" is used twice in the text but

there is no precise mention of the actual data structure used to

represent the Bloom filter/CBF.

- in 3.2 "unlike previous studies in the literature" add citations

Reviewer #3: This paper presents ntStat, a software toolkit for characterizing raw sequencing data through $k$-mer statistics. Its contribution is twofold. First, it provides a $k$-mer counting module that uses Bloom filters to achieve a balance of speed, low memory usage, and minimal disk footprint compared to existing tools. An innovation of this module is its ability to concurrently track not only the total frequency of each $k$-mer but also its "depth", defined as the number of distinct sequencing reads in which it appears. Second, the paper introduces a histogram analysis module that uses evolutionary algorithms to model the distribution of $k$-mer counts. This approach adapts to data from various sequencing platforms, including high-error long-read technologies, to provide de novo estimates of genomic characteristics such as heterozygosity, sequencing quality, and $k$-mer coverage.

The paper is well-written and clear, however, the presented results make it somewhat difficult to fully grasp the specific problems that ntStat is designed to solve.

Concerning the histogram analysis, an in-depth comparison between the capabilities and performance of GenomeScope2 and those of ntStat would be beneficial for the manuscript's readability.

A similar concern appears regarding the $k$-mer counting implementation. The implementation uses ideas very similar to BFCounter, so a summary of the key differences between the methods seems necessary. Other efficient $k$-mer counting tools exist, for instance, Gerbil, which is very efficient for large $k$ (Gerbil: a fast and memory-efficient $k$-mer counter with GPU-support), or hackgap (Fast Gapped $k$-mer Counting with Subdivided Multi-Way Bucketed Cuckoo Hash Tables), which follows a similar in-memory approach but with Cuckoo filters. Additionally, the widely used and efficient, though unpublished, tool FASTK should be considered. As a result, the current benchmark is not truly representative of the state of the art. Furthermore, it is somewhat unclear in which situations ntStat actually shines.

Finally, although the notion of $k$-mer depth is interesting, a discussion regarding the utility of this metric and its differences from classic $k$-mer abundance seems necessary. Moreover, using a Bloom filter to avoid counting duplicate $k$-mers within a given read seems computationally expensive and can introduce false positives, as two distinct $k$-mers could be mapped to the same hash value. A hash table would likely be more efficient and would avoid this unnecessary supplementary bias.

**Have the authors made all data and (if applicable) computational code underlying the findings in their manuscript fully available?**

The PLOS Data policy requires authors to make all data and code underlying the findings described in their manuscript fully available without restriction, with rare exception (please refer to the Data Availability Statement in the manuscript PDF file). The data and code should be provided as part of the manuscript or its supporting information, or deposited to a public repository. For example, in addition to summary statistics, the data points behind means, medians and variance measures should be available. If there are restrictions on publicly sharing data or code —e.g. participant privacy or use of data from a third party—those must be specified.requires authors to make all data and code underlying the findings described in their manuscript fully available without restriction, with rare exception (please refer to the Data Availability Statement in the manuscript PDF file). The data and code should be provided as part of the manuscript or its supporting information, or deposited to a public repository. For example, in addition to summary statistics, the data points behind means, medians and variance measures should be available. If there are restrictions on publicly sharing data or code —e.g. participant privacy or use of data from a third party—those must be specified.requires authors to make all data and code underlying the findings described in their manuscript fully available without restriction, with rare exception (please refer to the Data Availability Statement in the manuscript PDF file). The data and code should be provided as part of the manuscript or its supporting information, or deposited to a public repository. For example, in addition to summary statistics, the data points behind means, medians and variance measures should be available. If there are restrictions on publicly sharing data or code —e.g. participant privacy or use of data from a third party—those must be specified.requires authors to make all data and code underlying the findings described in their manuscript fully available without restriction, with rare exception (please refer to the Data Availability Statement in the manuscript PDF file). The data and code should be provided as part of the manuscript or its supporting information, or deposited to a public repository. For example, in addition to summary statistics, the data points behind means, medians and variance measures should be available. If there are restrictions on publicly sharing data or code —e.g. participant privacy or use of data from a third party—those must be specified.

Reviewer #1: **No:**  the simulated data as well as the code to reproduce the result are not available. the ntStat package is available.the simulated data as well as the code to reproduce the result are not available. the ntStat package is available.the simulated data as well as the code to reproduce the result are not available. the ntStat package is available.the simulated data as well as the code to reproduce the result are not available. the ntStat package is available.

Reviewer #2: None

Reviewer #3: Yes

PLOS authors have the option to publish the peer review history of their article (what does this mean?). If published, this will include your full peer review and any attached files.). If published, this will include your full peer review and any attached files.). If published, this will include your full peer review and any attached files.). If published, this will include your full peer review and any attached files.

...

Reviewer #1: No

Reviewer #2: No

Reviewer #3: No

**Figure resubmission:**

**Reproducibility:**



---

## [Decision Letter · Decision Letter 1]

19 Mar 2026

PCOMPBIOL-D-25-01236R1

ntStat: k-mer characterization using occurrence statistics in raw sequencing data

PLOS Computational Biology

Dear Dr. Kazemi,

Thank you for submitting your manuscript to PLOS Computational Biology. After careful consideration, we feel that it has merit but does not fully meet PLOS Computational Biology's publication criteria as it currently stands. Therefore, we invite you to submit a revised version of the manuscript that addresses the points raised during the review process.

We look forward to receiving your revised manuscript.

Kind regards,

Michael Domaratzki

Academic Editor

PLOS Computational Biology

Nir Ben-Tal

Section Editor

PLOS Computational Biology

**Additional Editor Comments:**

I understand that the revision to the sections was done to fit the journal's guidelines. I would recommend that reviewer 1's comments on the discussions/conclusions be considered in the context of whether the total information in the former discussion and conclusion sections is in the newest version.

**Journal Requirements:**

Please provide an Author Summary. This should appear in your manuscript between the Abstract (if applicable) and the Introduction, and should be 150-200 words long. The aim should be to make your findings accessible to a wide audience that includes both scientists and non-scientists. Sample summaries can be found on our website under Submission Guidelines:

**Reviewers' comments:**

Reviewer's Responses to Questions

**Comments to the Authors:**

Reviewer #1: The authors have addressed most of my concerns. After reading the responses and review comments from other reviewers, I find the histogram analysis part meant for applications in genome assembly field, which I have no experience.

The conclusion and discussion are deleted in the revised manuscript. I think it is necessary to keep the discussion/conclusion such that general readers can understand what are the typical applications of this software and what are the difference compared with other software like GenomeScope.

The paper can be published after putting back the discussions/conclusions.

Reviewer #2: The authors have adequately addressed my comments regarding the data structure and the modularity of the tool. They have also added benchmark curves, which improves the evaluation of the method. In addition, they provided more details about the k-mer size and clarified the difference with GenomeScope.

Note: the sentence “NtStat is designed to model the underlying probability of individual k-mers belonging to error, heterozygous or homozygous components” might be better placed in Section 2.2, particularly in relation to another reviewer’s comment noting that the paper does not clearly explain the motivation behind the histogram.

Reviewer #3: I have re-reviewed the revised version of the manuscript describing ntStat, a toolkit for characterizing raw sequencing data through k-mer statistics, combining an efficient k-mer counting strategy that tracks both k-mer frequency and the “depth” (number of distinct reads containing the k-mer), and a histogram analysis module based on evolutionary algorithms to infer de novo genomic and sequencing characteristics across a range of sequencing technologies, including error-prone long-read data.

The authors have addressed my previous concerns in a thorough and constructive manner. The revised manuscript now more clearly articulates the distinct scope and aims of ntStat compared to GenomeScope, notably by emphasizing ntStat’s ability to provide k-mer-level statistics (e.g., error probability of individual k-mers) rather than only global genome-wide estimates. The manuscript also better differentiates the k-mer counting component from related approaches such as BFCounter by explicitly highlighting ntStat’s thresholding capabilities, depth extraction, and spaced seed support. Importantly, the benchmarking section has been strengthened by the inclusion of hackgap, which is particularly relevant given the shared support for spaced seeds, improving the representativeness of the comparison.

In addition, the discussion of k-mer depth has been expanded and is now substantially more convincing, especially regarding its utility for characterizing repetitive elements via the count-to-depth ratio. The authors have also clarified the rationale behind the read-level Bloom filter design, including its sizing to make false positives effectively negligible in this context and the practical benefits of fixed-size memory use and reuse of precomputed hashes.

Overall, I am satisfied with the current state of the manuscript and believe that the revisions have improved clarity, positioning with respect to existing tools, and the motivation for the proposed metrics and design choices. I thank the authors for their diligent work in addressing the review comments.

Sincerely,

**Have the authors made all data and (if applicable) computational code underlying the findings in their manuscript fully available?**

The PLOS Data policy requires authors to make all data and code underlying the findings described in their manuscript fully available without restriction, with rare exception (please refer to the Data Availability Statement in the manuscript PDF file). The data and code should be provided as part of the manuscript or its supporting information, or deposited to a public repository. For example, in addition to summary statistics, the data points behind means, medians and variance measures should be available. If there are restrictions on publicly sharing data or code —e.g. participant privacy or use of data from a third party—those must be specified.requires authors to make all data and code underlying the findings described in their manuscript fully available without restriction, with rare exception (please refer to the Data Availability Statement in the manuscript PDF file). The data and code should be provided as part of the manuscript or its supporting information, or deposited to a public repository. For example, in addition to summary statistics, the data points behind means, medians and variance measures should be available. If there are restrictions on publicly sharing data or code —e.g. participant privacy or use of data from a third party—those must be specified.requires authors to make all data and code underlying the findings described in their manuscript fully available without restriction, with rare exception (please refer to the Data Availability Statement in the manuscript PDF file). The data and code should be provided as part of the manuscript or its supporting information, or deposited to a public repository. For example, in addition to summary statistics, the data points behind means, medians and variance measures should be available. If there are restrictions on publicly sharing data or code —e.g. participant privacy or use of data from a third party—those must be specified.requires authors to make all data and code underlying the findings described in their manuscript fully available without restriction, with rare exception (please refer to the Data Availability Statement in the manuscript PDF file). The data and code should be provided as part of the manuscript or its supporting information, or deposited to a public repository. For example, in addition to summary statistics, the data points behind means, medians and variance measures should be available. If there are restrictions on publicly sharing data or code —e.g. participant privacy or use of data from a third party—those must be specified.

Reviewer #1: Yes

Reviewer #2: Yes

Reviewer #3: Yes

PLOS authors have the option to publish the peer review history of their article (what does this mean?). If published, this will include your full peer review and any attached files.). If published, this will include your full peer review and any attached files.). If published, this will include your full peer review and any attached files.). If published, this will include your full peer review and any attached files.

...

Reviewer #1: No

Reviewer #2: No

Reviewer #3: No

**Figure resubmission:**
---

## [Editor Report · Decision Letter 2]

23 Mar 2026

Dear Mr. Kazemi,

We are pleased to inform you that your manuscript 'ntStat: k-mer characterization using occurrence statistics in raw sequencing data' has been provisionally accepted for publication in PLOS Computational Biology.

Best regards,

Michael Domaratzki

Academic Editor

PLOS Computational Biology

Nir Ben-Tal

Section Editor

PLOS Computational Biology

---

## [Editor Report · Acceptance letter]

PCOMPBIOL-D-25-01236R2

ntStat: k-mer characterization using occurrence statistics in raw sequencing data

Dear Dr Kazemi,

I am pleased to inform you that your manuscript has been formally accepted for publication in PLOS Computational Biology. Your manuscript is now with our production department and you will be notified of the publication date in due course.

With kind regards,

Judit Kozma
